# MODEL-BASED OFFLINE REINFORCEMENT LEARNING WITH CONSERVATIVE BIDIRECTIONAL ROLLOUTS

## ABSTRACT

Offline reinforcement learning (offline RL) learns from an offline dataset without further interactions with the environment. Although such offline training patterns can avoid cost and damage in the real environment, one main challenge is the distributional shift between the state-action pairs visited by the learned policy and those in the offline dataset. Prevailed existing model-based offline RL approaches learn a dynamics model from the dataset and perform pessimistic policy optimization based on uncertainty estimation. However, the inaccurate quantification of model uncertainty may incur the poor generalization and performance of model-based approaches, especially in the datasets lacking of sample diversity. To tackle this limitation, we instead design a novel framework for model-based offline RL, named **C**onservative **O**ffline **Bi**directional **M**odel-based Policy **O**ptimization (abbr. as COBiMO). First, we learn an ensemble bidirectional model from the offline dataset and construct long bidirectional rollouts by joining two unidirectional ones, thereby increasing the diversity of the model rollouts. Second, we devise a conservative rollout method that minimizes the reconstruction loss, further improving the sample accuracy. We theoretically prove that the bound of rollout error of COBiMO is tighter than the ones using the unidirectional models. Empirical results also show that COBiMO outperforms previous offline RL algorithms on the widely used benchmark *D4RL*.

## 1 INTRODUCTION

In typical reinforcement learning (RL), the objective is to train an agent through real-time interactions with the environment and the consequent reward signals (Sutton & Barto, 2018). In recent years, such trial-and-error learning pattern has exhibited its significant superiority in complex decision-making tasks, e.g. robotic locomotion skill learning (Mnih et al., 2013; Andrychowicz et al., 2020), go game (Silver et al., 2017; Schrittwieser et al., 2020), drone racing (Kaufmann et al., 2023). However, the trial-and-error of typical RL can be impractical in real-world settings. Especially for those cost-sensitive tasks, the tremendous interactions required for online training may cause various intricated issues, e.g. crashing cars in autonomous driving (Zhou et al., 2023) and extremely expensive data collection in healthcare (Yu et al., 2021a; Qayyum et al., 2020).

As a potential alternative to this problem, offline reinforcement learning (offline RL), also known as batch RL, has attracted great attention (Fujimoto et al., 2019; Lange et al., 2012; Kumar et al., 2019; Agarwal et al., 2020; Peng et al., 2019; Wu et al., 2019). In an offline setting, one only has access to a static dataset which is pre-collected based on past experiences, and any further interactions with the environment are forbidden throughout the entire training process. Therefore, the offline learning mode can avoid damage and cost in the real environment caused by intermediate policies. Moreover, even when online interactions are feasible, learning from offline datasets still holds the edge if the dynamics are sophisticated and hard to generalize (Levine et al., 2020).

However, offline RL algorithms suffer severely from *distributional shift* caused by the mismatch between the behavior policy used for generating the dataset and the learned policy (Levine et al., 2020; Kostrikov et al., 2021). During the training process, the agent may encounter some state-action pairs that rarely or even never appear in the dataset. In the following policy evaluation stage, such out-of-distribution (shorted as OOD) actions tend to be overestimated with spurious high Q-values, leading to poor performance when tested or deployed in the real environment. To handle with

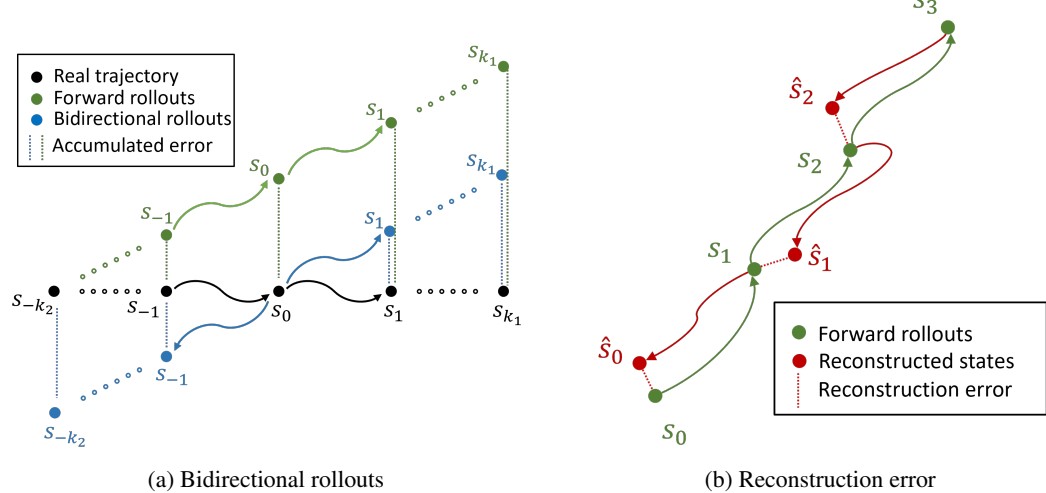

(a) Bidirectional rollouts          (b) Reconstruction error

Figure 1: Conservative bidirectional rollouts in a snapshot

distributional shift, prevailing model-free offline RL approaches tend to conservatively ensure that the distribution over actions under the learned policy is close to the behavior distribution (Levine et al., 2020). Such restrictions can be enforced by directly adding a hard policy constraint in the policy update step (Kumar et al., 2019; Fujimoto et al., 2019; Ghasemipour et al., 2021; Siegel et al., 2020; Nair et al., 2020), or implicitly incorporating some form of penalty into the Q-value function towards OOD state-action pairs (Kumar et al., 2020; Kostrikov et al., 2021; Wu et al., 2019; Nachum et al., 2019). Another branch of model-free algorithms aims to quantify the epistemic uncertainty of the Q-value functions and then utilizes uncertainty estimates to produce conservative target values (Lakshminarayanan et al., 2017; Bai et al., 2022; An et al., 2021).

While model-free offline RL directly optimizes the policy with the pre-collected experiences, model-based offline RL additionally learns a dynamics model from the experiences in the dataset. Mainstreams of existing model-based offline RL approaches incorporate conservatism based on the model uncertainty, by adding a penalty either on the learned reward function (Yu et al., 2020; Chen et al., 2021; Kidambi et al., 2020) or on the Q-function during policy optimization (Yu et al., 2021b; Wang et al., 2021; Yue et al., 2023). By generating and training on the additional imaginary rollouts, model-based methods have the ability for higher sample efficiency and broader generalization, therefore alleviating distributional shifts.

However, a key factor that plagues model-based approaches is the inaccurate quantification of uncertainty (Yu et al., 2021b; Bai et al., 2022), leading to the low quality of the model rollouts. Also, model-based methods tend to perform poorly under scenarios that lack sample diversity, e.g. datasets collected by a single behavior policy (Yu et al., 2020). Obviously, data quality is directly dependent on the model accuracy and the design of the rollout method. In addition, the rollout length, i.e. the number of steps to generate a trajectory, affects both the data quality and diversity. Intuitively, generating long trajectories produces more diverse imaginary data while accumulating error upper bound since further model rollouts need to build on the previous imaginary transitions.

Motivated by the above intuition, we propose **C**onservative **O**ffline **Bi**directional **M**odel-based Policy **O**ptimization (abbr. as COBiMO), a novel model-based framework for offline RL. For each direction, i.e. forward and backward, COBiMO learns a dynamics model along with a rollout policy based on the offline dataset. As Figure 1a shows, we divide an imaginary trajectory generation into two parts: the forward rollouts and the backward rollouts. By constructing a long trajectory with two shorter ones, COBiMO can increase the diversity of the generated samples while maintaining the data quality. Moreover, for each rollout part, we apply a conservative rollout method that aims to minimize the summed reconstruction loss (Figure 1b). We provide a theoretical analysis for the effectiveness of COBiMO in Section 4. COBiMO presents a plug-and-pull data augmentation to be combined with existing model-free offline RL algorithms. Extensive experiments on the widely used benchmark *D4RL* (Fu et al., 2020) demonstrate the superiority of our method.

## 2 PRELIMINARIES

### 2.1 MARKOV DECISION PROCESS

Markov Decision Process (MDP) can be described as a tuple $(\mathcal{S}, \mathcal{A}, T, r, \gamma, \mu_0)$ (Sutton & Barto, 2018). Precisely, $\mathcal{S}$ and $\mathcal{A}$ denote the state and action set respectively; $T : \mathcal{S} \times \mathcal{S} \times \mathcal{A} \to \mathbb{R}$ is the transition distribution function embedded in the real dynamics, where $T(s'|s, a)$ denotes the probability of transiting to the next state $s'$ assuming action $a$ is taken in current state $s$; $r(s, a) :$ $\mathcal{S} \times \mathcal{A} \to \mathbb{R}$ is the reward function; $\gamma \in (0, 1)$ is the discount factor that determines the importance of future rewards compared to immediate ones; $\mu_0(s) : \mathcal{S} \to [0, 1]$ is the initial state distribution.

The goal of RL is to seek the optimal policy $\pi(a|s) : \mathcal{S} \times \mathcal{A}$ which maximizes the expected cumulative discounted returns:

$$\mathcal{J}(\pi) := \mathbb{E}_{s_0 \sim \mu_0, a_t \sim \pi(\cdot|s_t), s_{t+1} \sim T(\cdot|s_t, a_t)} \left[ \sum_{t=0}^{\infty} \gamma^t r(s_t, a_t) \right] \tag{1}$$

### 2.2 MODEL-BASED OFFLINE REINFORCEMENT LEARNING

In the offline RL setting, the agent only has access to a static dataset $\mathcal{D} = \{(s, a, r, s')\}$ and any further interactions with the environment are not allowed. The offline dataset can be generated through one or multi-source behavior policies. We denote the empirical distribution of behavior policy induced by the dataset as $\pi_\mathcal{D}$. In most existing offline RL works, $\pi_\mathcal{D}$ is not accessible.

Model-based RL approaches aim at performing planning or policy search based on a learned model of the environment. Typically, a forward dynamics model $\hat{T}(s'|s, a)$ along with the reward model $\hat{r}(s, a)$ is learned through pre-collected experiences. For the simplicity of notations, we incorporate the reward model into the dynamics model, denoted by $\hat{T}(s'|s, a)$ or $\hat{T}(s', r|s, a)$. In this paper, we design a bidirectional framework which contains a forward dynamics model $\hat{T}_f(s'|s, a)$ and a backward dynamics model $\hat{T}_b(s|s', a)$.

## 3 METHODOLOGY

In the offline RL setting, we only have access to a pre-collected dataset $\mathcal{D}$. We assume that there exists a forward transition dynamics $T_f$ and a behavior policy $\beta$ induced by $\mathcal{D}$. In this section, we first introduce the components involved in COBiMO and then explain how these modules are organically combined to conservatively generate bidirectional model rollouts.

### 3.1 COMPONENTS OF COBiMO

Before stepping into the overall framework of COBiMO , we first introduce the four components involved: a forward dynamics model $\hat{T}_f$, a backward dynamics model $\hat{T}_b$, a forward rollout policy $\hat{\pi}_f$ and a backward rollout policy $\hat{\pi}_b$.

**The forward and backward dynamics model.** As is done in MBPO (Janner et al., 2019), we model the forward transition dynamics $T_f$ with a neural network parameterized by $\theta_f$ and $\phi_f$, outputting a Gaussian distribution $\hat{T}_f$ over the next state and the corresponding reward: $\hat{T}_f(s', r|s, a) = \mathcal{N}(\mu_{\theta_f}(s, a), \Sigma_{\phi_f}(s, a))$. In practical implementation, we learn an ensemble of $N$ forward models $\{\hat{T}_f^i = \mathcal{N}(\mu_{\theta_f^i}, \Sigma_{\phi_f^i})\}_{i=1}^N$ and only use $K$ elite ones. We train the forward model to maximize the log-likelihood function, i.e. to minimize the forward model loss function $\mathcal{L}_{fm}(\theta_f, \phi_f)$:

$$\mathcal{L}_{fm}(\theta_f, \phi_f) := \mathbb{E}_{(s, a, s', r) \sim \mathcal{D}} \left[ -\log \hat{T}_f(s', r|s, a) \right] \tag{2}$$

For the backward model, we employ the same model design as the forward one and aim to approximate the backward transition dynamics $T_b$, which is defined by $T_b(s, r|s', a) = T_f(s', r|s, a)$. Therefore, we train each backward model $\hat{T}_b(s, r|s', a) = \mathcal{N}(\mu_{\theta_b}(s', a), \Sigma_{\phi_b}(s', a))$ to minimize

the backward model loss function $\mathcal{L}_{bm}(\theta_b, \phi_b)$:

$$\mathcal{L}_{bm}(\theta_b, \phi_b) := \mathbb{E}_{(s,a,s',r)\sim\mathcal{D}}\left[-\log \hat{T}_b(s, r|s', a)\right] \tag{3}$$

**The forward and backward rollout policy.** With the learned forward and backward models, COBiMO can generate one-step imaginary transition $(s, a, r, s')$ given the query $(s, a)$ or $(s', a)$. To roll out a multi-step trajectory $(s_0, a_0, \hat{s}_1, \hat{a}_1, \ldots, \hat{s}_T, \hat{a}_T)$, a sequence of imaginary actions $\{\hat{a}_i\}_{i=1}^T$ are indispensable. Therefore, we additionally learn a rollout policy for each unidirectional model.

To approximate the behavior policy $\beta$ induced by the dataset $\mathcal{D}$, we learn a forward rollout policy $\pi_f$ by behavioral cloning. Following BCQ (Fujimoto et al., 2019), we learn a generative model $\hat{G}_f^\omega = (\hat{E}_f^{\omega_1}, \hat{D}_f^{\omega_2})$ using a Conditional Variational Auto-encoder (shorted as CVAE) (Sohn et al., 2015). The encoder $\hat{E}_f^{\omega_1}(s, a)$ represents a gaussian distribution given the state-action pair $(s, a)$ and outputs a latent vector $z \sim \hat{E}_f^{\omega_1}(s, a)$; the decoder $\hat{D}_f^{\omega_2}$ takes the state s and the latent vector $z$ as the input and reconstructs the input action $\tilde{a} = \hat{D}_f^{\omega_2}(s, z)$.

We train the forward generative model $\hat{G}_f^\omega(s)$ by minimizing the CVAE loss $\mathcal{L}_{fp}$,

$$\mathcal{L}_{fp}(\hat{G}_f^\omega) = \mathbb{E}_{(s,a,s',r)\sim\mathcal{D}, z\sim\hat{E}_f^{\omega_1}(s,a)}\left[(a - \hat{D}_f^{\omega_2}(s, z))^2 + D_{KL}\left(\hat{E}_f^{\omega_1}(s, a)\|\mathcal{N}(0, I)\right)\right] \tag{4}$$

where the first term inside the expectation is the reconstruction loss and the second term stands for an additional penalty for the KL-divergence between $\hat{E}_f^{\omega_1}$ and the multivariate standard Gaussian distribution $\mathcal{N}(0, I)$.

As for the backward rollout policy, we train another CVAE $\hat{G}_b^\xi = (\hat{E}_b^{\xi_1}, \hat{D}_b^{\xi_2})$ that takes the next state $s'$ as the input. Similarly, we train the backward generative model $\hat{G}_b^\xi$ by minimizing the loss function $\mathcal{L}_{bp}$:

$$\mathcal{L}_{bp}(\hat{G}_b^\xi) = \mathbb{E}_{(s,a,s',r)\sim\mathcal{D}, z\sim\hat{E}_b^{\xi_1}(s',a)}\left[(a - \hat{D}_b^{\xi_2}(s', z))^2 + D_{KL}\left(\hat{E}_b^{\xi_1}(s', a)\|\mathcal{N}(0, I)\right)\right] \tag{5}$$

By drawing the latent vector $z \sim \mathcal{N}(0, I)$ instead, we can generate actions from CVAEs given a fixed state. For simplicity, we respectively use $\hat{\pi}_f$ and $\hat{\pi}_b$ to represent the policies induced by the model $\hat{G}_f^\omega$ and $\hat{G}_b^\xi$. Combining the dynamics models with the rollout policies, we can generate rollouts of any length and any direction from a given state $s_0$:

**Definition 3.1** (Forward model rollouts). *We say that $\tau_f = \{(s_i, a_i, s_{i+1}, r_i)\}_{i=0}^{N-1}$ are forward model rollouts of length $N$ starting at $s_0$ if the following conditions are satisfied:*

$$a_i \sim \hat{\pi}_f(\cdot|s_i), (s_{i+1}, r_i) \sim \hat{T}_f(\cdot|s_i, a_i), i \in \{0, 1, \ldots, N-1\} \tag{6}$$

Similarly, we can define backward model rollouts $\tau_b = \{(s_i, a_i, s_{i+1}, r_i)\}_{i=-N}^{-1}$. We defer the formal definition in the Appendix A.

## 3.2 FRAMEWORK OF COBiMO

To generate an imaginary trajectory of length $k$, one straightforward idea is to roll out $k$ steps from the given state in a fixed direction, iteratively using the rollout policy to pick actions and the dynamics model to transit to the next states. Apparently, larger trajectory length increases the diversity of the generated transitions, but the error caused by the inaccuracy of the dynamics model will accumulate. In this subsection, we will introduce the conservative bidirectional rollout method that improves the accuracy of the imaginary rollouts even if the trajectories are multi-step.

**Conservative bidirectional rollouts.** As Figure 1a shows, we divide a trajectory of length $k$ into two shorter ones: a forward trajectory of length $k_1$ and a backward trajectory of length $k_2$ satisfying $k_1 + k_2 = k$. Specifically, given a starting state $s_0$, we use model $\hat{T}_f$ and rollout policy $\hat{\pi}_f$ to generate forward model rollouts of length $k_1$, and use $\hat{T}_b$ and $\hat{\pi}_b$ to backtrack $k_2$ steps. By joining the forward and backward rollouts from the same starting state $s_0$, we get a trajectory $\tau = \{(s_i, a_i, s_{i+1}, r_i)\}_{i=-k_2}^{k_1-1}$ of length $k = k_1 + k_2$.

Note that COBiMO learns the forward and backward dynamics models to respectively approximate $T$ and $T^{-1}$. Given a forward model rollout, we assume that if we back-construct it with the learned backward model and rollout policy from the ending state, the discrepancy between the forward trajectory and the reconstructed backward one should be small, as illustrated in Figure 1b. Following this intuition, we design a conservative rollout method to reduce the reconstruction error:

**Definition 3.2** (Reconstruction error). *Given forward rollouts $\tau_f = \{(s_i, a_i, s_{i+1}, r_i)\}_{i=0}^{N-1}$ starting at $s_0$ and ending at $s_N$, the reconstruction error using backward model $\hat{T}_b$ is defined as follow.*

$$\mathcal{E}_r(\tau_f) := \frac{1}{N} \sum_{i=0}^{N-1} \mathbb{E}_{\hat{s}_i \sim \hat{T}_b(\cdot|s_{i+1}, a_i)} \left[ \|s_i - \hat{s}_i\| \right] \tag{7}$$

Similarly, we can define the reconstruction error for backward rollouts using forward model $\hat{T}_f$, which can be found in Appendix A. For each direction, we generate certain number of candidate rollouts and compute the reconstruction loss for each candidate. Only the one with the least reconstruction error is selected and added to the model replay buffer $\mathcal{B}$.

**Overall framework.** To sum up, the framework of COBiMO is shown in Algorithm 1. We first learn two dynamics models $\hat{T}_f, \hat{T}_b$ and two rollout policies $\pi_f, \pi_b$ using the experiences from the dataset $\mathcal{D}$. Unlike the Dyna-style approaches (Sutton & Barto, 2018; Lai et al., 2020), these components stay unchanged during the subsequent procedures once the training is finished. Then we combine these four modules to produce a model replay buffer $\mathcal{B}$ by generating conservative bidirectional rollouts. Adding $\mathcal{B}$ to the dataset $\mathcal{D}$, we can extend the original dataset and run any model-free offline RL method on it to learn a target policy. We provide the implementation details along with pseudo-codes involved in this section in Appendix C.

---

**Algorithm 1** Conservative Offline Bidirectional Model-based Policy Optimization (COBiMO)

---

**Input:** Offline dataset $\mathcal{D}$; forward rollout step $k_1$; backward rollout step $k_2$; learning rates $\alpha_\theta, \alpha_\phi, \alpha_\omega, \alpha_\xi$ ; the number of iterations $T_m, T_p$; the size $N$ of model replay buffer $\mathcal{B}$.

1: Randomly initialize model parameters $\theta_f, \phi_f, \theta_b, \phi_b$.
2: **for** $T_m$ epochs **do**              ▷ Learn the forward and backward dynamics model.
3:     Compute forward model loss $\mathcal{L}_{fm}$ by Equation 2.
4:     Update $\theta_f \leftarrow \theta_f - \alpha_\theta \nabla_{\theta_f} \mathcal{L}_{fm}$.
5:     Update $\phi_f \leftarrow \phi_f - \alpha_\phi \nabla_{\phi_f} \mathcal{L}_{fm}$.
6:     Compute backward model loss $\mathcal{L}_{bm}$ by Equation 3.
7:     Update $\theta_b \leftarrow \theta_b - \alpha_\theta \nabla_{\theta_b} \mathcal{L}_{bm}$.
8:     Update $\phi_b \leftarrow \phi_b - \alpha_\phi \nabla_{\phi_b} \mathcal{L}_{bm}$.
9: **end for**
10: Randomly initialize rollout policy parameters $\omega, \xi$.
11: **for** $T_p$ epochs **do**              ▷ Learn the forward and backward rollout policy.
12:     Compute forward rollout policy loss $\mathcal{L}_{fp}$ by Equation 4.
13:     Update $\omega \leftarrow \omega - \alpha_\omega \nabla_\omega \mathcal{L}_{fp}$.
14:     Compute backward rollout policy loss $\mathcal{L}_{bp}$ by Equation 5.
15:     Update $\xi \leftarrow \xi - \alpha_\xi \nabla_\xi \mathcal{L}_{bp}$.
16: **end for**
17: Initialize the model replay buffer $\mathcal{B} \leftarrow \emptyset$.
18: **while** $|\mathcal{B}| < N$ **do**                      ▷ Generate model rollouts.
19:     Sample the starting state $s_0$ from $\mathcal{D}$.
20:     Generate conservative bidirectional model rollouts $\tau = \{(s_i, a_i, s_{i+1}, r_i)\}_{i=-k_2}^{k_1-1}$ using dynamics models $\hat{T}_f, \hat{T}_b$ and rollout policies $\hat{G}_f^\omega, \hat{G}_b^\xi$.
21:     Add model rollouts to the model replay buffer $\mathcal{B} \leftarrow \mathcal{B} \cup \tau$.
22: **end while**
23: Add model replay buffer to the dataset $\mathcal{D} \leftarrow \mathcal{D} \cup \mathcal{B}$.
24: Run any model-free offline RL algorithm (e.g. CQL) on $\mathcal{D}$ to learn the final policy $\pi$.
**Output:** The learned policy $\pi$.

---

## 4 THEORETICAL ANALYSIS

### 4.1 NOTATIONS

Before stepping into the theoretical analysis of COBiMO , we first introduce the involved notations briefly. As previously stated, $\hat{T}_f$ and $\hat{T}_b$ stand for the learned forward and backward dynamics model, $\hat{\pi}_f$ and $\hat{\pi}_b$ represent the learned rollout policies. $\epsilon_{fm}, \epsilon_{bm}, \epsilon_{fp}$ and $\epsilon_{bm}$ respectively represent the error of these 4 learned components. We denote the state marginal of forward model at time $t$ by $\hat{T}_f^t(s)$ and backward one by $\hat{T}_b^{-t}(s)$. The total variation distance between two distributions $p_1$ and $p_2$ is denoted by $D_{TV}(p_1, p_2)$. The definitions for these notations can be found in Appendix A.

### 4.2 PERFORMANCE GUARANTEE FOR COBiMO

**Lemma 4.1** (Recursive error for forward state marginal). *Suppose the error for forward dynamics model is $\epsilon_{fm}$ and the error for forward rollout policy is $\epsilon_{fp}$. Then we can bound the error of state marginal for forward model at time $t + 1$ as follows:*

$$D_{TV}(\hat{T}_f^{t+1}, T_f^{t+1}) \leq D_{TV}(\hat{T}_f^t, T_f^t) + \epsilon_{fm} + \epsilon_{fp} \tag{8}$$

Along with Lemma B.2, Lemma 4.1 indicates that the state marginal error of model rollouts at each timestep can be attributed to the one at previous timestep plus the error of model and rollout policy. Therefore, the error of rollouts will accumulate since further imaginary states depend on the previous ones, coinciding with the intuition from Figure 1a.

**Definition 4.2** (Cumulative error for imaginary rollouts). *We define the cumulative error of an imaginary trajectory $\tau = \{(s_i, a_i, s_{i+1}, r_i)\}_{i=0}^{N-1}$ based on the total variation distance:*

$$\mathcal{E}_{TV}(\tau) := \sum_{t=0}^{N-1} \tau^t D_{TV}(\hat{T}^t(s_t, a_t), T^t(s_t, a_t)), \tag{9}$$

*where $\hat{T}^t(s_t, a_t)$ is the state-action joint distribution at time $t$ that generates $(s_t, a_t)$. $T^t(s_t, a_t)$ is the true joint distribution.*

The above definition provides a reasonable measure that evaluates the quality of an imaginary trajectory by quantifying the cumulative discounted discrepancy between the true joint distribution and the one that generates the rollouts. We involve the discount factor $\tau$ to be consistent with the definition of cumulative discounted returns $\mathcal{J}$.

As described before, we can either use the unidirectional models to directly generate a long trajectory or employ the bidirectional model to construct one with two shorter parts. We can derive the upper bound for the cumulative error of both rollout methods:

**Theorem 4.3.** *Assume that $\tau = 1$, $\epsilon_m = \epsilon_{fm} = \epsilon_{bm}$ and $\epsilon_p = \epsilon_{fp} = \epsilon_{bp}$. Suppose that $\tau_f$ and $\tau_b$ are forward and backward rollouts of length $k_1 + k_2$, and $\hat{\tau}$ is the bidirectional rollouts composed of $k_1$ forward steps and $k_2$ backward steps. Then we have the upper bound for the cumulative error of each rollouts as follows:*

$$\mathcal{E}_{TV}(\tau_f) \leq \frac{(k_1 + k_2 + 1)(k_1 + k_2)}{2}(\epsilon_m + \epsilon_p) + (k_1 + k_2 + 1)\epsilon_p, \tag{10}$$

$$\mathcal{E}_{TV}(\tau_b) \leq \frac{(k_1 + k_2 + 1)(k_1 + k_2)}{2}(\epsilon_m + \epsilon_p) + (k_1 + k_2 + 1)\epsilon_p, \tag{11}$$

$$\mathcal{E}_{TV}(\hat{\tau}) \leq \frac{(k_1 + 1)k_1 + (k_2 + 1)k_2}{2}(\epsilon_m + \epsilon_p) + (k_1 + k_2 + 1)\epsilon_p, \tag{12}$$

Theorem 4.3 provides the upper bounds for the cumulative error of unidirectional model rollouts and bidirectional ones of the same length. By comparing the two bounds, we can clearly conclude that COBiMO can obtain a tighter bound of the cumulative error than the unidirectional models when generating multi-step trajectories. Formal proof for Theorem 4.3 can be found in Appendix B.

## 5 EXPERIMENTS

To evaluate our proposed algorithm, we conduct extensive experiments on *D4RL* (Fu et al., 2020), a widely-used benchmark for offline RL. We demonstrate the effectiveness of COBiMO by answering the following questions:

- **Q1:** How does COBiMO perform on the offline RL benchmark compared to both model-free and model-based offline RL baselines?
- **Q2:** How does the proposed conservative bidirectional rollout method affect COBiMO?
- **Q3:** How does COBiMO perform when plugged into other model-free methods?

We conduct extensive experiments in three environments under the benchmark *D4RL*, i.e. Maze2D, AntMaze, and MuJoCo. We provide a snapshot of the environments in Appendix D. Each environment has various dataset types as follows:

- *Maze2D*: The Maze2D domain tries to navigate a ball agent along a 2D maze and gains rewards by reaching a target goal location. There are 3 possible maze layouts (i.e., *umaze*, *medium*, and *large*).
- *AntMaze*: In the AntMaze domain, the ball is replaced with an "Ant" quadruped robot. Therefore, the AntMaze domain is much more challenging than Maze2D since it additionally involves a low-level locomotion problem. Three flavors of datasets are introduced, (1) *fixed*: the ant needs to reach a specific goal from a fixed location, (2) *play*: the ant is commanded to specific hand-picked locations (which are not necessarily the goal at evaluation), starting from a different set of hand-picked start locations, (3) *diverse*: the start location and the goal are randomly initialized.
- *Gym-MuJoCo*: Three locomotion environments are involved in Gym-MuJoCo, i.e. *Hopper*, *Walker2D* and *Halfcheetah*. In each environment, there are four dataset types,(1) experiences in the *random* dataset are collected by a random policy, (2) the *medium* dataset is generated by an early-stopping SAC policy, (3) the *medium-replay* dataset is composed of the *medium* dataset plus all records in the replay buffer during the early-stopping training process, (4) the *medium-expert* dataset is mixed up with equal amounts of optimal experiences and suboptimal ones.

### 5.1 COMPARISON ANALYSIS

We compare our method with algorithms from three domain:

- Imitation learning: behavior cloning (BC) learned from the dataset;
- Model-free offline RL methods: BCQ (Fujimoto et al., 2019), BEAR (Kumar et al., 2019), AWR (Peng et al., 2019), CQL (Kumar et al., 2020);
- Model-based offline RL methods: RepB-SDE (Lee et al., 2020), MOPO (Yu et al., 2020), MAPLE (Chen et al., 2021).

We present the overall results in Table 1. As proposed in (Fu et al., 2020), each number is a normalized score of the policy at the last iteration of training, averaged over 6 random seeds. We build COBiMO on CQL and conduct five runs on different random seeds to get an average score. For the sake of reproducibility, a detailed statement about the architectures and hyper-parameters is presented in Appendix C. We will release our code after anonymous cancellation.

Among the involved model-free and model-based methods, we find that COBiMO achieves the highest normalized score in 6 out of all 12 tasks. Moreover, COBiMO performs competitively compared with the best one in 3 out of the 6 remaining datasets (hopper-random, hopper-medium-expert, halfcheetah-medium-replay), validating the superiority of our method.

COBiMO beats CQL in most datasets, showing that COBiMO is a powerful framework that can be easily plugged into existing model-free offline RL approaches. COBiMO outperforms other baselines in all *medium* datasets. This coincides with our motivation that COBiMO can generalize well even in the domain of less diversity.

Table 1: Results of COBiMO and other baselines on Gym-MuJoCo datasets. We take results of MOPO, MAPLE and RepB-SDE from their original papers; numbers for other model-free algorithms are reported from the *D4RL* benchmark paper (Fu et al., 2020). The highest scores across all algorithms are bolded.

| Dataset type | BC | BCQ | BEAR | AWR | CQL | RepB-SDE | MOPO | MAPLE | COBiMO |
|---|---|---|---|---|---|---|---|---|---|
| walker2d-random | 1.6 | 4.9 | 7.3 | 1.5 | 7.0 | 21.1 | 13.6 | **21.7** | 11.8 |
| walker2d-medium | 6.6 | 53.1 | 59.1 | 17.4 | 79.2 | 72.1 | 17.8 | 56.3 | **80.6** |
| walker2d-medium-replay | 11.3 | 15.0 | 19.2 | 15.5 | 26.7 | 49.8 | 39.0 | **76.7** | 41.3 |
| walker2d-medium-expert | 6.4 | 57.5 | 40.1 | 53.8 | 111.0 | 88.8 | 44.6 | 73.8 | **113.5** |
| hopper-random | 9.8 | 10.6 | 11.4 | 10.2 | 10.8 | 8.6 | **11.7** | 10.6 | 9.4 |
| hopper-medium | 29.0 | 54.5 | 52.1 | 35.9 | 58.0 | 34.0 | 28.0 | 21.1 | **75.3** |
| hopper-medium-replay | 11.8 | 33.1 | 33.7 | 28.4 | 48.6 | 62.2 | 67.5 | **87.5** | 54.3 |
| hopper-medium-expert | **111.9** | 110.9 | 96.3 | 27.1 | 98.7 | 82.6 | 23.7 | 42.5 | 99.3 |
| halfcheetah-random | 2.1 | 2.2 | 25.1 | 2.5 | 35.4 | 32.9 | 35.4 | 38.4 | **42.9** |
| halfcheetah-medium | 36.1 | 40.7 | 41.7 | 37.4 | 44.4 | 49.1 | 42.3 | 50.4 | **51.3** |
| halfcheetah-medium-replay | 38.4 | 38.2 | 38.6 | 40.3 | 46.2 | 57.5 | 53.1 | **59.0** | 55.3 |
| halfcheetah-medium-expert | 35.8 | 64.7 | 53.4 | 52.7 | 62.4 | 55.4 | 63.3 | 63.5 | **65.4** |

## 5.2 ABLATION ANALYSIS

In order to answer Q2, we design three variants of COBiMO as follows.

- BiMO: COBiMO w/o conservative rollout method which minimizes reconstruction error.
- COBiMO-fwd: COBiMO w/o forward model rollouts.
- COBiMO-bwd: COBiMO w/o backward model rollouts.

Comparison of COBiMO and the above three ablation methods can be found in Table 2. BiMO outperforms two other unidirectional variants by a significant margin, indicating that the bidirectional builds are likely to be more effective than the conservative rollout method. Also, COBiMO-fwd performs slightly better than COBiMO-bwd, consistent with Wang et al. (2021) that the backward model can be trusted just as (or even more than) the traditional forward model.

Generally, the ablation experiments confirm that both the bidirectional setting and the conservative rollout method are vital to the performance of COBiMO.

Table 2: Results of COBiMO and three ablation methods on Gym-MuJoCo datasets.

| Dataset type | COBiMO | BiMO | COBiMO-fwd | COBiMO-bwd |
|---|---|---|---|---|
| walker2d-random | **11.8** | 8.5 | 6.1 | 5.2 |
| walker2d-medium | **80.6** | 76.3 | 68.6 | 70.1 |
| walker2d-medium-replay | **41.3** | 33.2 | 37.6 | 35.9 |
| walker2d-medium-expert | **113.5** | 98.1 | 77.3 | 75.2 |
| hopper-random | 9.4 | 8.2 | **10.2** | 7.3 |
| hopper-medium | **75.3** | 65.3 | 50.5 | 38.4 |
| hopper-medium-replay | **54.3** | 35.5 | 42.8 | 32.7 |
| hopper-medium-expert | **99.3** | 87.5 | 60.2 | 69.1 |
| halfcheetah-random | **42.9** | 36.1 | 26.7 | 31.8 |
| halfcheetah-medium | **51.3** | 30.7 | 25.9 | 23.6 |
| halfcheetah-medium-replay | **55.3** | 47.4 | 35.2 | 32.4 |
| halfcheetah-medium-expert | **65.4** | 55.2 | 44.2 | 47.1 |

## 5.3 ADAPTABILITY ANALYSIS

To demonstrate the adaptability of COBiMO, we apply it to two other model-free offline RL algorithms, BCQ and BEAR. The results are shown in Table 3. We run COBiMO-BCQ and COBiMO-BEAR on five different random seeds and calculate the average scores. Both COBiMO-BCQ and COBiMO-BEAR defeat their original algorithms by a significant margin in 10 out of 12 datasets, suggesting that COBiMO provides robust data augmentation for different model-free RL methods.

Table 3: Normalized scores of COBiMO and the corresponding baselines on Maze2D and AntMaze datasets. We take results of BCQ and BEAR from the *D4RL* benchmark paper (Fu et al., 2020). We bold the higher scores for each pair of original ones and our method.

| Dataset type | BCQ | COBiMO-BCQ | BEAR | COBiMO-BEAR |
|---|---|---|---|---|
| maze2d-umaze | **12.8** | 6.1 | 3.4 | **7.3** |
| maze2d-medium | 8.3 | **57.5** | 29 | **45.9** |
| maze2d-large | 6.2 | **28.9** | 4.6 | **10.3** |
| antmaze-umaze-fixed | **78.9** | 32.4 | **73** | 59.8 |
| antmaze-umaze-diverse | 55 | **65.7** | 61 | **72.9** |
| antmaze-medium-play | 0 | **5.0** | 0 | **9.6** |
| antmaze-medium-diverse | 0 | **15.8** | 8 | **13.5** |
| antmaze-large-play | 6.7 | **14.3** | 0 | 0 |
| antmaze-large-diverse | 2.2 | **11.4** | 0 | **7.9** |

## 6 RELATED WORK

**Model-free offline RL.** Most model-free methods try to keep the target policy close to the behavior policy used to collect the offline dataset. BCQ (Fujimoto et al., 2019) learns the bahavior policy by a CVAE and applies perturbation on actions for diversity. BEAR (Kumar et al., 2019) restricts actions outputed by the learned policy within the support set of the training distribution. BRAC (Wu et al., 2019) provides a powerful framework that allows different behavior policy regularizers. AWR (Peng et al., 2019) builds on ideas from reward-weighted regression and is able to learn from the offline dataset. Kumar et al. (2020) proposes a strong algorithm named CQL, which learns a conservative Q-function such that the expected value of a policy under this Q-function lower-bounds its true value. Unlike these methods, COBiMO learns from the dataset as well as the model rollouts.

**Model-based offline RL.** Model-based approaches learn a dynamics model before or along with the policy optimization. Yu et al. (2020) proposes a framework based on MBPO (Janner et al., 2019) with reward penalty named MOPO. RepB-SDE (Lee et al., 2020) focuses on learning the representation for a robust model of the environment under the distributional shift. MAPLE (Chen et al., 2021) aims to learn a target policy that can adapt its behavior in out-of-support regions when deployed. COMBO (Yu et al., 2021b) incorporates conservatism by regularizing the value function on out-of-support state-action tuples generated via model rollouts. ROMI (Wang et al., 2021) learns a reverse dynamics model and a corresponding rollout policy to generate new transitions. Out of offline RL settings, Lai et al. (2020) propose BMPO, a bidirectional framework for typical RL. BMPO follows the Dyna-style algorithm(Wang et al., 2019) and trains the model iteratively together with the target policy, making it inapplicable to offline RL since ongoing interactions are needs. As far as we know, COBiMO is the first algorithm that applies bidirectional rollouts into offline RL.

## 7 CONCLUSION

In this work, we introduce a novel framework for model-based offline RL, namely Conservative Offline Bidirectional Model-based Policy Optimization (COBiMO). COBiMO adopts a conservative bidirectional rollout method for more accurate and diverse generalization. We theoretically prove a tighter bound of the rollout error of COBiMO than unidirectional ones. Empirical results on *D4RL* also demonstrates the superiority of our method.

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

## A    NOTATIONS AND DEFINITIONS

Here we provide the missing definitions of the notations involved in the main paper for better understanding.

**Definition A.1** (Backward model rollout). *We say that $\tau_b = \{(s_i, a_i, s_{i+1}, r_i)\}_{i=-k}^{-1}$ is a backward model rollout of length $k$ starting at $s_0$ if the following conditions are satisfied:*

$$a_i \sim \pi_b(s_{i+1}), (s_i, r_i) \sim \hat{T}_b(\cdot|s_{i+1}, a_i), i \in \{-1, -2, \ldots, -k\} \tag{13}$$

**Definition A.2** (Reconstruction error for backward rollout). *Given a backward rollout $\tau_b = \{(s_i, a_i, s_{i+1}, r_i)\}_{i=-N}^{-1}$ starting at $s_0$ and ending at $s_{-N}$, the reconstruction error using backward model $\hat{T}_f$ is defined as follow.*

$$\mathcal{E}(\tau_b) := \frac{1}{N} \sum_{i=-N}^{-1} \mathbb{E}_{\hat{s}_{i+1} \sim \hat{T}_f(\cdot|s_i, a_i)} [\|s_{i+1} - \hat{s}_{i+1}\|] \tag{14}$$

**Definition A.3** (Total variation distance). *Given two probability density functions $p$ and $q$, the total variation distance $D_{TV}$ is defined as:*

$$D_{TV}(p, q) := \frac{1}{2} \sum_x |p(x) - q(x)| \tag{15}$$

**Definition A.4** (State marginal for forward and backward model). *We denote the state marginal at time $t$ induced by forward model $\hat{T}_f$ and rollout policy $\hat{\pi}_f$ as $\hat{T}_f^t$:*

$$\hat{T}_f^t(s) := \mathbb{P}_{s_0 \sim \mathcal{D}, a_i \sim \hat{\pi}_f(s_i), s_t \sim \hat{T}_f(\cdot|s_i, a_i)} [s_t = s|s_0] \tag{16}$$

*Likewise, the state marginal for backward model is defined as:*

$$\hat{T}_b^{-t}(s) := \mathbb{P}_{s_0 \sim \mathcal{D}, a_i \sim \hat{\pi}_b(s_{i+1}), s_t \sim \hat{T}_b(\cdot|s_{i+1}, a_i)} [s_{-t} = s|s_0] \tag{17}$$

**Definition A.5.** *We define the upper bound $\epsilon_{fm}$ and $\epsilon_{bm}$ for the learned models $\hat{T}_f$ and $\hat{T}_b$ as:*

$$\epsilon_{fm} := \max_t \mathbb{E}_{(s,a) \sim \hat{T}_f^t} \left[ D_{TV}(\hat{T}_f(\cdot|s, a), T_f(\cdot|s, a)) \right] \tag{18}$$

$$\epsilon_{bm} := \max_t \mathbb{E}_{(s',a) \sim \hat{T}_b^t} \left[ D_{TV}(\hat{T}_b(\cdot|s', a), T_b(\cdot|s', a)) \right] \tag{19}$$

**Definition A.6** (Error for forward and backward rollout policy). *We define the upper bound $\epsilon_{fp}$ and $\epsilon_{bp}$ for the learned rollout policies $\hat{\pi}_f$ and $\hat{\pi}_b$ as:*

$$\epsilon_{fp} := \max_s D_{TV}(\hat{\pi}_f(\cdot|s), \beta(\cdot|s)) \tag{20}$$

$$\epsilon_{bp} := \max_{s'} D_{TV}(\hat{\pi}_b(\cdot|s'), \beta_b(\cdot|s')) \tag{21}$$

*where $\beta$ is the behavior policy and $\beta_b$ is the backward one.*

## B    PROOF FOR THEOREM 4.3

**Lemma B.1** (TVD of joint distributions, (Janner et al., 2019), Lemma B.1). *Suppose we have two joint distributions $p_1(x, y) = p_1(x)p_1(y|x)$ and $p_2(x, y) = p_2(x)p_2(y|x)$. We can bound the total variation distance as:*

$$D_{TV}(p_1(x, y), p_2(x, y)) \le D_{TV}(p_1(x), p_2(x)) + \max_x D_{TV}(p_1(y|x), p_2(y|x)) \tag{22}$$

Lemma B.1 provides a bound for the total variation distance between two joint distributions using their marginal distributions and conditional distributions.

Before we formally prove Theorem 4.3, we first restate Lemma 4.1 of a backward version and provide theoretical proof. The proof for Lemma 4.1 can be completed almost identically.

**Lemma B.2** (Recursive error for backward state marginal). *Suppose the error for the backward model is $\epsilon_{bm}$ and the error for the backward rollout policy is $\epsilon_{bp}$. Then we can bound the error of state marginal for the backward model at time $-(t+1)$ as follows:*

$$D_{TV}(\hat{T}_b^{-(t+1)}(s), T_b^{-(t+1)}(s)) \leq D_{TV}(\hat{T}_b^{-t}(s), T_b^{-t}(s)) + \epsilon_{bm} + \epsilon_{bp} \tag{23}$$

*Proof.*

$$\left| \hat{T}_b^{-(t+1)}(s) - T_b^{-(t+1)}(s) \right|$$

$$= \left| \sum_{s',a} \hat{T}_b^{-t}(s',a)\hat{T}_b(s|s',a) - \sum_{s',a} T_b^{-t}(s',a)T_b(s|s',a) \right|$$

$$= \left| \sum_{s',a} [\hat{T}_b^{-t}(s',a)\hat{T}_b(s|s',a) - T_b^{-t}(s',a)T_b(s|s',a)] \right|$$

$$\leq \sum_{s',a} \left| \hat{T}_b^{-t}(s',a)\hat{T}_b(s|s',a) - T_b^{-t}(s',a)T_b(s|s',a) \right|$$

$$= \sum_{s',a} \left| \hat{T}_b^{-t}(s',a)(\hat{T}_b(s|s',a) - T_b(s|s',a)) + (\hat{T}_b^{-t}(s',a) - T_b^{-t}(s',a))T_b(s|s',a) \right|$$

$$\leq \sum_{s',a} \hat{T}_b^{-t}(s',a) \left| \hat{T}_b(s|s',a) - T_b(s|s',a) \right| + \sum_{s',a} \left| \hat{T}_b^{-t}(s',a) - T_b^{-t}(s',a) \right| T_b(s|s',a)$$

Sum up the last equation over $s$ and then divide by 2. For the first item, we have

$$\frac{1}{2} \sum_s \sum_{s',a} \hat{T}_b^{-t}(s',a) \left| \hat{T}_b(s|s',a) - T_b(s|s',a) \right|$$

$$= \sum_s \mathbb{E}_{s',a \sim \hat{T}_b^{-t}} \left[ \frac{1}{2} | \hat{T}_b(s|s',a) - T_b(s|s',a) | \right]$$

$$= \mathbb{E}_{s',a \sim \hat{T}_b^{-t}} [D_{TV}(\hat{T}_b(\cdot|s',a), T_b(\cdot|s',a))]$$

$$\leq \epsilon_{bm}$$

For the second item, we have

$$\frac{1}{2} \sum_s \sum_{s',a} \left| \hat{T}_b^{-t}(s',a) - T_b^{-t}(s',a) \right| T_b(s|s',a)$$

$$= \frac{1}{2} \sum_{s',a} \left[ \left| \hat{T}_b^{-t}(s',a) - T_b^{-t}(s',a) \right| \left( \sum_s T_b(s|s',a) \right) \right]$$

$$= \frac{1}{2} \sum_{s',a} \left| \hat{T}_b^{-t}(s',a) - T_b^{-t}(s',a) \right|$$

$$= D_{TV}(\hat{T}_b^{-t}(s',a), T_b^{-t}(s',a)) \qquad \text{(Lemma B.1)}$$

$$\leq D_{TV}(\hat{T}_b^{-t}(s'), T_b^{-t}(s')) + \max_{s'} D_{TV}(\hat{\pi}_b(\cdot|s'), \beta_b(\cdot|s'))$$

$$= D_{TV}(\hat{T}_b^{-t}(s), T_b^{-t}(s)) + \epsilon_{bp}$$

Therefore, we have

$$D_{TV}(\hat{T}_b^{-(t+1)}(s), T_b^{-(t+1)}(s)) \leq D_{TV}(\hat{T}_b^{-t}(s), T_b^{-t}(s)) + \epsilon_{bm} + \epsilon_{bp}$$

$\square$

**Remark.** Lemma 4.1 and Lemma B.2 are straightforward to understand: the error for the next state marginal is composed of the previous one, the error of rollout policy and the error of the dynamic model, because generating the next state of the rollouts needs to continue on the previous one and then sample an action from the rollout policy and finally transit by the dynamic model.

**Lemma B.3.** *Suppose the joint distribution of the forward dynamic model $\hat{T}_f^t(s, a)$ starts at $t = 0$ and the one of the backward dynamic model starts at $t = T$. We can bound the total variation distance between the joint distributions between the dynamic model and the true ones as follows:*

$$D_{TV}(\hat{T}_f^t(s, a), T_f^t(s, a)) \leq t(\epsilon_{fm} + \epsilon_{fp}) + \epsilon_{fp}, \; t \geq 0 \tag{24}$$

$$D_{TV}(\hat{T}_b^t(s, a), T_b^t(s, a)) \leq (T - t)(\epsilon_{bm} + \epsilon_{bp}) + \epsilon_{bp}, \; t \leq T \tag{25}$$

*Proof.* By Lemma 4.1, Lemma B.2 and Lemma B.1, we have:

$$D_{TV}(\hat{T}_f^t(s, a), T_f^t(s, a)) \leq D_{TV}(\hat{T}_f^t(s), T_f^t(s)) + \max_s D_{TV}(\hat{\pi}_f(\cdot|s), \beta(\cdot|s)),$$

$$\leq D_{TV}(\hat{T}_f^t(s), T_f^t(s)) + \epsilon_{fp}$$

$$\leq t(\epsilon_{fm} + \epsilon_{fp}) + \epsilon_{fp}$$

Similarly,

$$D_{TV}(\hat{T}_b^t(s, a), T_b^t(s, a)) \leq D_{TV}(\hat{T}_b^t(s), T_b^t(s)) + \max_{s'} D_{TV}(\hat{\pi}_b(\cdot|s'), \beta_b(\cdot|s')),$$

$$\leq D_{TV}(\hat{T}_b^t(s), T_b^t(s)) + \epsilon_{bp}$$

$$\leq (T - t)(\epsilon_{bm} + \epsilon_{bp}) + \epsilon_{bp}$$

$\square$

Now we provide the theoretical proof for Theorem 4.3.

*Proof.* We first prove the bounds for discounted cumulative error. The undiscounted cumulative error can be easily derived via discounted ones.

Suppose that $\tau_f$ are forward model rollouts of length $k_1 + k_2$ starting at $s_0$, $\tau_b$ are backward model rollouts of length $k_1 + k_2$ starting at $s_{k_1+k_2}$, $\hat{\tau}$ are bidirectional model rollouts composed of $k_1$ forward steps and $k_2$ backward steps starting at $s_{k_2}$. Therefore, all these three rollouts span from $t = 0$ to $t = k_1 + k_2$.

By Definition 4.2, the (discounted) cumulative error of these rollouts are as follows:

$$\mathcal{E}_{TV}(\tau_f) = \sum_{t=0}^{k_1+k_2} \tau^t D_{TV}(\hat{T}_f^t(s_t, a_t), T_f^t(s_t, a_t)) \tag{26}$$

$$\mathcal{E}_{TV}(\tau_b) = \sum_{t=0}^{k_1+k_2} \tau^t D_{TV}(\hat{T}_b^t(s_t, a_t), T_b^t(s_t, a_t)) \tag{27}$$

$$\mathcal{E}_{TV}(\hat{\tau}) = \sum_{t=0}^{k_2} \tau^t D_{TV}(\hat{T}_b^t(s_t, a_t), T_b^t(s_t, a_t)) + \sum_{t=1}^{k_1} \tau^{k_2+t} D_{TV}(\hat{T}_f^t(s_t, a_t), T_f^t(s_t, a_t)) \tag{28}$$

Assume that $\epsilon_m = \epsilon_{fm} = \epsilon_{bm}$ and $\epsilon_p = \epsilon_{fp} = \epsilon_{bp}$. Applying Lemma B.3 to the above equations, we have:

$$\mathcal{E}_{TV}(\tau_f) \leq \sum_{t=0}^{k_1+k_2} \tau^t [t(\epsilon_m + \epsilon_p) + \epsilon_p] \tag{29}$$

$$\mathcal{E}_{TV}(\tau_b) \leq \sum_{t=0}^{k_1+k_2} \tau^t [(k_1 + k_2 - t)(\epsilon_m + \epsilon_p) + \epsilon_p] \tag{30}$$

$$\mathcal{E}_{TV}(\hat{\tau}) \leq \sum_{t=0}^{k_2} \tau^t [(k_2 - t)(\epsilon_m + \epsilon_p) + \epsilon_p] + \sum_{t=1}^{k_1} \tau^{k_2+t} [t(\epsilon_m + \epsilon_p) + \epsilon_p] \tag{31}$$

Sum up the series on the RHS of Equation (29)-(31) by simple mathematics:

$$\mathcal{E}_{TV}(\tau_f) \leq \frac{\tau - (k_1 + k_2 + 1)\tau^{k_1+k_2+1} + (k_1 + k_2)\tau^{k_1+k_2+2}}{(1-\tau)^2}(\epsilon_m + \epsilon_p) + \frac{1 - \tau^{k_1+k_2+1}}{1-\tau}\epsilon_p \tag{32}$$

$$\mathcal{E}_{TV}(\tau_b) \leq \frac{k_1 + k_2 - (k_1 + k_2 + 1)\tau + \tau^{k_1+k_2+1}}{(1-\tau)^2}(\epsilon_m + \epsilon_p) + \frac{1 - \tau^{k_1+k_2+1}}{1-\tau}\epsilon_p \tag{33}$$

$$\mathcal{E}_{TV}(\hat{\tau}) \leq \frac{k_2 - (k_2 + 1)\tau + 2\tau^{k_2+1} - (k_1 + 1)\tau^{k_1+k_2+1} + k_1\tau^{k_1+k_2+2}}{(1-\tau)^2}(\epsilon_m + \epsilon_p)$$
$$+ \frac{1 - \tau^{k_1+k_2+1}}{1-\tau}\epsilon_p \tag{34}$$

So far, we obtain the bounds for discounted cumulative error. To achieve the bounds of Theorem 4.3, simply set $\tau = 1$ and use L'Hôpital's rule twice:

$$\mathcal{E}_{TV}(\tau_f) \leq \frac{(k_1 + k_2 + 1)(k_1 + k_2)}{2}(\epsilon_m + \epsilon_p) + (k_1 + k_2 + 1)\epsilon_p, \tag{35}$$

$$\mathcal{E}_{TV}(\tau_b) \leq \frac{(k_1 + k_2 + 1)(k_1 + k_2)}{2}(\epsilon_m + \epsilon_p) + (k_1 + k_2 + 1)\epsilon_p, \tag{36}$$

$$\mathcal{E}_{TV}(\hat{\tau}) \leq \frac{(k_1 + 1)k_1 + (k_2 + 1)k_2}{2}(\epsilon_m + \epsilon_p) + (k_1 + k_2 + 1)\epsilon_p, \tag{37}$$

$\square$

**Remark.** Divide the above inequalities by $k_1 + k_2$, we can easily find the the bound of $\mathcal{E}_{TV}(\hat{\tau})$ is much tighter than the other one, indicating that COBiMO can obtain a tighter bound of the cumulative error than the uni-directional model rollouts.

## C   IMPLEMENTATION DETAILS

In this section, we provide all pseudo-codes and implementation details involved in Section 3.

### C.1   DYNAMIC MODELS

---
**Algorithm 2** Leaning dynamic models.

---
**Input:** Offline dataset $\mathcal{D}$; learning rates $\alpha_\theta, \alpha_\phi$ ; the number of iterations $T_m$; the ensemble number $N$; the elite number $K$; the holdout ratio $h$.
1: Randomly divide $\mathcal{D}$ into training set $\mathcal{D}_{train}$ and holdout set $\mathcal{D}_{holdout}$ by the holdout ratio $h$.
2: Randomly initialize parameters $\{\theta_f^i, \phi_f^i, \theta_b^i, \phi_b^i\}_{i=1}^N$ for each individual model.
3: **for** $T_m$ epochs **do**
4:    **for** $i = 1, \ldots, N$ **do**
5:       Compute loss $\mathcal{L}_{fm}^i$ for forward model $i$ on $\mathcal{D}_{train}$ by Equation 2.
6:       Update $\theta_f^i \leftarrow \theta_f^i - \alpha_\theta \nabla_{\theta_f^i} \mathcal{L}_{fm}^i$
7:       Update $\phi_f^i \leftarrow \phi_f^i - \alpha_\phi \nabla_{\phi_f^i} \mathcal{L}_{fm}^i$.
8:       Compute loss $\mathcal{L}_{bm}^i$ for backward model $i$ on $\mathcal{D}_{train}$ by Equation 3.
9:       Update $\theta_b^i \leftarrow \theta_b^i - \alpha_\theta \nabla_{\theta_b^i} \mathcal{L}_{bm}^i$
10:      Update $\phi_b^i \leftarrow \phi_b^i - \alpha_\phi \nabla_{\phi_b^i} \mathcal{L}_{bm}^i$.
11:   **end for**
12: **end for**
13: Compute model error for each forward and backward model based on $\mathcal{D}_{rollout}$ by Equation 38 and 39. Select $K$ elite ones out of $N$ models with the least model error.
**Output:** The elite ensemble models $\{\mathcal{N}(\mu_{\theta_f^i}, \Sigma_{\phi_f^i})\}_{i=1}^K$ and $\{\mathcal{N}(\mu_{\theta_b^i}, \Sigma_{\phi_b^i})\}_{i=1}^K$.

---

**Definition C.1** (Model Error). *Given the dataset $\mathcal{D}$ and the distribution $T_D$ induced by $\mathcal{D}$ , the expected error of the forward dynamic model $\hat{T}_f$ is defined as:*

$$\epsilon_D(\hat{T}_f) = \mathbb{E}_{(s,a,s',r)\sim\mathcal{D},\hat{s}'\sim\hat{T}_f(\cdot|s,a)}\left[\|\hat{s}' - s'\|^2\right] \tag{38}$$

*Similarly, we can define the expected error of the backward dynamic model $\hat{T}_b$ as:*

$$\epsilon_D(\hat{T}_b) = \mathbb{E}_{(s,a,s',r)\sim\mathcal{D},\hat{s}\sim\hat{T}_b(\cdot|s',a)}\left[\|\hat{s} - s\|^2\right] \tag{39}$$

The detailed training framework for models can be found in Algorithm 2. Following Yu et al. (2020) and Wang et al. (2021), we set the ensemble number $N = 7$ and elite number $K = 5$. Each model in the ensemble is represented as a 4-layer feedforward neural network with 200 hidden units. The holdout ratio is 0.2 as most works do. The learning rates $\alpha_\theta$ and $\alpha_\phi$ are set to 0.001. For model rollouts, we randomly pick one from the elite models.

## C.2 ROLLOUT POLICIES.

---
**Algorithm 3** Learning rollout policies.

---
**Input:** Offline dataset $\mathcal{D}$; learning rates $\alpha_\omega, \alpha_\sigma$ ; the number of iterations $T_p$; batch size $N_B$.
1: Randomly initialize parameters $\omega, \sigma$ for CVAE $\hat{G}_f^\omega, \hat{G}_b^\xi$.
2: **for** $T_p$ epochs **do**
3:     Initialize $L_{fp} \leftarrow 0$, $L_{bp} \leftarrow 0$.
4:     Sample a mini-batch of $N_B$ transitions $(s, a, s', r)$ from $\mathcal{D}$.
5:     **for** $N_B$ transitions **do**
6:         Sample $(\mu_f, \sigma_f)$ from $E_f^{\omega_1}(s, a)$, $z$ from $\mathcal{N}(\mu_f, \sigma_f)$ and $\tilde{a}$ from $D_f^{\omega_2}(s, z)$.
7:         Sum the loss $L_{fp} \leftarrow L_{fp} + (a - \tilde{a}) + D_{KL}(\mathcal{N}(\mu_f, \sigma_f)\|\mathcal{N}(0, I))$.
8:         Sample $(\mu_b, \sigma_b)$ from $E_b^{\xi_1}(s', a)$, $z$ from $\mathcal{N}(\mu_b, \sigma_b)$ and $\tilde{a}$ from $D_b^{\xi_2}(s', z)$
9:         Sum the loss $L_{fp} \leftarrow L_{fp} + (a - \tilde{a}) + D_{KL}(\mathcal{N}(\mu_b, \sigma_b)\|\mathcal{N}(0, I))$
10:     **end for**
11:     Update $\omega \leftarrow \omega - \alpha_\omega \nabla_\omega \mathcal{L}_{fp}$
12:     Update $\xi \leftarrow \xi - \alpha_\xi \nabla_\xi \mathcal{L}_{bp}$
13: **end for**
**Output:** The rollout policies $\hat{\pi}_f = \hat{G}_f^\omega$ and $\hat{\pi}_b = \hat{G}_b^\xi$.

---

The rollout policies are learned as Algorithm 3. As for the involved hyperparameters, we set $\alpha_\omega = \alpha_\xi = 0.001$; batch size $N_B = 100$.

## C.3 CONSERVATIVE BIDIRECTIONAL ROLLOUTS

---
**Algorithm 4** Conservative bidirectional rollouts.

---
**Input:** offline dataset $\mathcal{D}$; forward rollout step $k_1$; backward rollout step $k_2$; dynamic models $\hat{T}_f, \hat{T}_b$; rollout policies $\hat{\pi}_f, \hat{\pi}_b$; the candidate number $C$.
1: Sample the initial state $s_0$ from $\mathcal{D}$.
2: **for** $i = 1, \ldots, C$ **do**
3:     Construct forward rollouts $\tau_f^i$ of $k_1$ steps by Definition 3.1.
4:     Compute the reconstruction loss of $\tau_f^i$ by Equation 7.
5:     Construct backward rollouts $\tau_b^i$ of $k_2$ steps by Definition A.1.
6:     Compute the reconstruction loss of $\tau_b^i$ by Equation 14.
7: **end for**
8: Select the best forward rollouts $\tau_f^*$ and backward rollouts $\tau_b^*$ with the least reconstruction error.
9: Construct an imaginary trajectory $\tau^*$ by joining $\tau_f^*$ and $\tau_b^*$ at the state $s_0$.
**Output:** Rollouts $\tau^*$.

---

We show the practical implementation of conservative bidirectional rollouts in Algorithm 4. Since the rollouts are generated by an ensemble bidirectional model, we set a longer rollout length than

other model-based offline RL methods (Yu et al., 2020; Wang et al., 2021; Yu et al., 2021b): $k_1 = k_2 = 1$ for *walker2d* environment and $k_1 = k_2 = 5$ for other environments. As for the candidate number, a larger $C$ provides more candidate rollouts to select but brings more computational costs. We set $C = 5$ for our implementation.

## D    EXPERIMENTAL SETTINGS AND RAW SCORES

### D.1    *D4RL* BENCHMARK

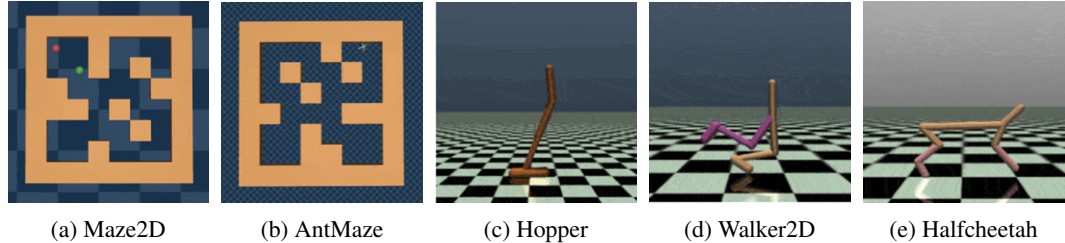

| (a) Maze2D | (b) AntMaze | (c) Hopper | (d) Walker2D | (e) Halfcheetah |

Figure 2: Environments in D4RL

We provide a snapshot of the *D4RL* environments in Figure 2

### D.2    RAW SCORES

As is proposed in (Fu et al., 2020), the experimental results in Section 5 are normalized to roughly lie between 0 and 100, where a score of 0 corresponds to the average returns of an agent taking actions uniformly at random across the action space and a score of 100 corresponds to the average returns of a domain-specific expert. Normalization is defined by the following equation:

$$S_{norm} = 100 \cdot \frac{S_{raw} - \text{random score}}{\text{expert score} - \text{random score}}$$

where $S_{norm}$ is the normalized score of the raw score $S_{raw}$.

We provide the raw scores of Table 1 and 2 in Table 4. In *Gym-MuJoCo* datasets, expert scores are taken from a soft-actor critic agent.

Table 4: The raw, un-normalized scores of COBiMO and other algorithms on Gym-MuJoCo datasets. Numbers for model-free algorithms are reported from the *D4RL* benchmark paper (Fu et al., 2020).

| Dataset type | SAC | BC | BCQ | BEAR | AWR | CQL | COBiMO | BiMO | COBiMO-fwd | COBiMO-bwd |
|---|---|---|---|---|---|---|---|---|---|---|
| walker2d-random | 4592.3 | 73.0 | 228.0 | 336.3 | 71.5 | 322.9 | 541.5 | 389.9 | 279.7 | 238.3 |
| walker2d-medium | 4592.3 | 304.8 | 2441.0 | 2717.0 | 800.7 | 2664.2 | 3701.7 | 3504.4 | 3150.9 | 3219.7 |
| walker2d-medium-replay | 4592.3 | 518.6 | 688.7 | 883.8 | 712.5 | 1227.3 | 1896.4 | 1524.4 | 1726.5 | 1648.4 |
| walker2d-medium-expert | 4592.3 | 297.0 | 2640.3 | 1842.7 | 2469.7 | 5097.3 | 5211.8 | 4505.1 | 3550.6 | 3454.2 |
| hopper-random | 3234.3 | 299.4 | 323.9 | 349.9 | 312.4 | 331.2 | 286.4 | 247.3 | 312.4 | 218.1 |
| hopper-medium | 3234.3 | 923.5 | 1752.4 | 1674.5 | 1149.5 | 2557.3 | 2430.4 | 2104.9 | 1623.2 | 1229.4 |
| hopper-medium-replay | 3234.3 | 364.4 | 1057.8 | 1076.8 | 904.0 | 1227.3 | 1747.3 | 1135.6 | 1373.1 | 1044.5 |
| hopper-medium-expert | 3234.3 | 3621.2 | 3588.5 | 3113.5 | 862.0 | 3192.0 | 3211.5 | 2827.9 | 1940.3 | 2229.7 |
| halfcheetah-random | 12135.0 | -17.9 | -1.3 | 2831.4 | 36.3 | 4114.8 | 5046.8 | 4202.7 | 3035.8 | 3668.9 |
| halfcheetah-medium | 12135.0 | 4196.4 | 4767.9 | 4897.0 | 4366.1 | 5232.1 | 6084.8 | 3525.5 | 2929.2 | 2643.5 |
| halfcheetah-medium-replay | 12135.0 | 4492.1 | 4463.9 | 4517.9 | 4727.4 | 5455.6 | 6588.9 | 5608.8 | 4095.1 | 3747.7 |
| halfcheetah-medium-expert | 12135.0 | 4169.4 | 7750.8 | 6349.6 | 6267.3 | 7466.9 | 7842.0 | 6576.5 | 5211.6 | 5571.4 |

The raw scores of Table 3 are listed in Table 5. For Maze2D, the expert is a hand-designed controller used to collect data. For AntMaze, the expert score equals 1, which is an estimate of the maximum score possible.

Table 5: The raw, un-normalized scores of COBiMO and the corresponding baselines on Maze2D and AntMaze datasets. We take numbers of BCQ and BEAR from the *D4RL* benchmark paper (Fu et al., 2020).

| Dataset type | BCQ | COBiMO-BCQ | BEAR | COBiMO-BEAR |
|---|---|---|---|---|
| maze2d-umaze | 41.5 | 32.2 | 28.6 | 34.0 |
| maze2d-medium | 35.0 | 165.0 | 89.8 | 134.4 |
| maze2d-large | 23.2 | 84.0 | 19.0 | 34.2 |
| antmaze-umaze-fixed | 0.8 | 0.3 | 0.7 | 0.6 |
| antmaze-umaze-diverse | 0.6 | 0.7 | 0.6 | 0.73 |
| antmaze-medium-play | 0.0 | 0.1 | 0.0 | 0.01 |
| antmaze-medium-diverse | 0.0 | 0.2 | 0.1 | 13.5 |
| antmaze-large-play | 0.1 | 0.1 | 0.0 | 0.0 |
| antmaze-large-diverse | 0.0 | 0.1 | 0.0 | 7.9 |

