# OpenReview forum: "Model-Based Offline Reinforcement Learning with Conservative Bidirectional Rollouts"
_ICLR.cc/2024/Conference — ICLR 2024 Conference Withdrawn Submission_

### Official Review · Reviewer_MDsd · 2023-10-26

**Soundness:** 2 fair
**Presentation:** 3 good
**Contribution:** 1 poor
**Rating:** 3
**Confidence:** 5

**Summary:**

Offline reinforcement learning (RL) suffers from the extrapolation error. There are numerous model-free and model-based offline RL algorithms that aim to tackle this challenge. Among them, model-based offline RL algorithms often learn a dynamics model from the dataset and perform pessimistic policy optimization based on uncertainty estimation. However, such quantifications are often inaccurate. This paper addresses this issue by training bidirectional dynamics models and rollout policies, and design a conservative rollout method that selects those synthetic transitions with the smallest reconstruction loss. The authors provide some theoretical analysis of their method and build their method upon some off-the-shelf model-free offline RL algorithms.

**Strengths:**

# Strengths

The strengths can be summarized below:

- this paper is well-motivated, and the whole paper structure is clear

- the logic flow of this paper is clear, and it is easy to follow and understand

- the authors provide theoretical analysis to support their method

**Weaknesses:**

# Weaknesses

Despite the aforementioned strengths, this paper has some flaws in novelty, empirical evaluation, and theoretical analysis. Based on these considerations, I can confirm that this paper is clearly under the acceptance bar of this venue. Please see the detailed comments below.

- (major) The core idea presented in this paper is NOT new. A highly relevant paper is published previously [x]. In [x], the authors also train bidirectional dynamics models and bidirectional rollout policies for offline data augmentation. Thus, the technical parts of this paper have a huge overlap with [x], making the contribution and significance of this paper quite weak. The differences are, that this paper selects the transitions with reconstruction loss while [x] selects reliable transitions via the proposed double check mechanism. It is doubtable whether the data selection approach adopted in this paper is better than the double check method, as intuitively, the reconstruction loss may not be reliable for forward/backward horizon larger than 1 (where no true next/previous states are available)

[x] Double Check Your State Before Trusting It: Confidence-Aware Bidirectional Offline Model-Based Imagination. NeurIPS 2022.

- (major) The empirical evaluations are limited and somewhat weak. The baseline algorithms this paper adopts are very old. It is somewhat confusing why the authors only choose to compare against these very weak algorithms. More advanced and recent offline RL algorithms ought to be included as the baselines (e.g., TD3BC, IQL, Decision Transformer, LAPO, etc.). The authors build their method upon CQL, BCQ, and BEAR. Can your method benefit more advanced offline RL algorithms?

- (major) This paper does not consider statistical significance. Written statements and the presentation of the results as tables (often without standard deviations) obscure this flaw. In fact, ALL tables in this paper does not include any signal of statistical significance, e.g., std, IQM. We have reached a point of maturity in the field where claims need to be made in reference to actual statistical evidence, which seems to be lacking in the current presentation.

- (major) The theoretical analysis is also not new. Similar techniques are adopted in the MBPO paper. Specifically, one online model-based RL algorithm BMPO [y] theoretically shows that the error of the bidirectional models is smaller than unidirectional models, making the theoretical insights of this paper less appealing and unsurprising.

[y] Bidirectional model-based policy optimization. ICML 2020.

- (minor) The authors ought to specify the version of the D4RL datasets they use in the paper. In Table 1, your evaluated scores in halfcheetah-medium-expert are questionably low, why is that?

- (minor) This paper does not do a good job in the related work part, the authors include too few recent offline model-based/model-free offline RL papers

**Questions:**

Please refer to the the weaknesses part.

---

### Official Review · Reviewer_qiBS · 2023-10-29

**Soundness:** 3 good
**Presentation:** 3 good
**Contribution:** 2 fair
**Rating:** 5
**Confidence:** 4

**Summary:**

This paper presents a new model-based method for offline reinforcement learning. The key technical contributions of the proposed model include: 1) It learns the bidirectional rollouts of the state transitions and the reward functions; 2) It learns forward and backward offline policies, following the BCQ method. With the learned bidirectional dynamics model and the corresponding policies, given a pivotal data point drawn from the offline dataset, the replay buffer can be augmented with the generated data trajectories.

Additionally, the paper provides a theoretical analysis, establishing a tighter bound on the rollout error for the conservative bidirectional rollouts compared to unidirectional approaches.

Finally, the empirical findings on the D4RL benchmark demonstrate the effectiveness of the proposed method.

**Strengths:**

1. The proposed method is simple, reasonable, and effective on the existing D4RL benchmark, showing great potential for practical offline RL applications.
2. The paper is well-written and easy to follow. The overall design of the proposed method is presented in a clear and thoroughly motivated manner.
3. The method seems to be a highly versatile framework. As shown in the paper, it can be easily integrated with existing model-free offline RL approaches.

**Weaknesses:**

1. My primary concern with this paper is about the novelty of the proposed bidirectional rollout technique. At NeurIPS 2022, a paper titled "Double Check Your State Before Trusting It: Confidence-Aware Bidirectional Offline Model-Based Imagination" by Lyu et al. introduces a conceptually similar idea. In both papers, forward and backward models are trained to augment the offline dataset. It is crucial for the authors to address this similarity and provide a comprehensive comparison between COBiMO and the method presented by Lyu et al., considering aspects such as model design and empirical results.
2. In the experiment section, the authors present averaged results of 6 random seeds. To enhance the statistical robustness of their findings, it would be better to include the standard deviations over multiple runs in Tables 1-3.
3. The paper primarily compares COBiMO with approaches that were proposed 2-3 years ago. It would be beneficial for the authors to extend their comparisons to include more recent advances in offline RL to provide a comprehensive evaluation of COBiMO's performance in the context of the most current state of the field.
4. In Section 5.3, there is an absence of an explanation regarding the factors that lead to performance degradation in certain tasks when COBiMO is applied (which can be reasonable but needs more analysis). Besides, as claimed in Section 5.3, the proposed method outperforms the original algorithms significantly in 10/12 tasks. However, it's essential to ensure that all relevant results supporting this claim are presented, as only a partial subset of the results is currently shown in Table 3.
5. Typos:
- In the first paragraph of Section 5.1, "...from three domain" should be corrected to "...from three domains".
- In the third paragraph of page 4, "...represents a gaussian distribution..." should be "...represents a Gaussian distribution...".

**Questions:**

In summary, my primary concerns include the technical novelty in comparison to the missing reference (major), and some finer details of the provided experimental results (minor).

---

### Official Review · Reviewer_7BFv · 2023-11-01

**Soundness:** 2 fair
**Presentation:** 3 good
**Contribution:** 1 poor
**Rating:** 3
**Confidence:** 4

**Summary:**

This paper studies the model-based offline reinforcement learning problem. The authors propose to learn bidirectional model and bidirectional behavioral policies and use them to generate rollout trajectories. The output policy is obtained by a model-free offline reinforcement learning on the augmented dataset. The paper provides theory and empirical study to justify the proposed algorithm.

**Strengths:**

1. The paper is clearly written and easy to follow.

**Weaknesses:**

1. The Related Work misses important paper. For instance, this paper is not the first to use bidirectional model in offline learning. Confidence-aware Bidirectional Offline Model-based Imagination is the first to apply this idea to the best of my knowledge.
2. I cannot recognize the algorithmic novelty of the algorithm. Forward imagination is widely used in model-based offline learning and Reverse Imagination was first proposed in ROMI. This paper seems to just combine these two ideas directly without justifying why it can substantially improve the performance
3. The theory seems to be trivial.
4. The experiment misses important baselines, such as ROMI and Confidence-aware Bidirectional Offline Model-based Imagination which share similar ideas. Besides, the performance does not seem compelling if one also look at the performance in ROMI and Confidence-aware Bidirectional Offline Model-based Imagination paper.

**Questions:**

1. What is the main intuition behind using bidirectional imagination? Why should we expect it provide substantial improvement?
2. What does the theory part tell us, is there any interesting insight?
3. How does the algorithm perform compared to other later model-based algorithms? How does the algorithm perform on other tasks in D4RL?

**Details Of Ethics Concerns:**

Not applicable.